# Coumarinolignoid and Indole Alkaloids from the Roots of the Hybrid Plant *Citrus × paradisi* Macfad (Rutaceae)

**DOI:** 10.3390/molecules28031078

**Published:** 2023-01-20

**Authors:** Fanny-Aimée Essombe Malolo, Ariane Dolly Kenmogne Kouam, Judith Caroline Ngo Nyobe, Lidwine Ngah, Marcel Frese, Jean Claude Ndom, Moses K. Langat, Bruno Ndjakou Lenta, Dulcie A. Mulholland, Norbert Sewald, Jean Duplex Wansi

**Affiliations:** 1Department of Chemistry, Faculty of Sciences, University of Douala, Douala P.O. Box 24157, Cameroon; 2Department of Pharmaceutical Sciences, Faculty of Medicine and Pharmaceutical Sciences, University of Douala, Douala P.O. Box 2701, Cameroon; 3Organic and Bioorganic Chemistry, Department of Chemistry, Bielefeld University, 33501 Bielefeld, Germany; 4Royal Botanic Gardens, Kew, Richmond TW9 3AE, UK; 5Department of Chemistry, Higher Teacher Training College, University of Yaoundé 1, Yaoundé P.O. Box 47, Cameroon; 6Natural Products Research Group, Department of Chemistry, Faculty of Engineering and Physical Sciences, University of Surrey, Guildford GU2 7XH, UK

**Keywords:** *Citrus × paradisi*, coumarinolignoid, indole alkaloids, chemotaxonomy, cytotoxicity

## Abstract

A phytochemical investigation of the roots of *Citrus × paradisi* Macfad. (Rutaceae) led to the isolation of two new compounds, namely 1-formyl-5-hydroxy-*N*-methylindolin-1-ium (**1**) and decyloxycleomiscosin D (**2**), along with ten known compounds: 1,1-dimethylpyrrolidin-1-ium-2-carboxylate (**3**), furan-2,3-diol (**4**), 5-methoxyseselin (**5**), umbelliferone (**6**), scopoletin (**7**), citracridone I (**8**), citracridone II (**9**), citracridone III (**10**), limonin (**11**) and lupeol (**12**). The structures were determined through the comprehensive spectroscopic analysis of 1D and 2D NMR and EI- and ESI-MS, as well as a comparison with the published data. Notably, compounds **3** and **4** from the genus *Citrus* are reported here for the first time. In addition, the MeOH extract of the roots and compounds **1**–**7** were screened against the human adenocarcinoma alveolar basal epithelial cell line A549 and the Caucasian prostate adenocarcinoma cell line PC3 using the MTT assay. While the extract showed significant activity, with IC_50_ values of 35.2 and 38.1 µg/mL, respectively, compounds **1**–**7** showed weak activity, with IC_50_ values of 99.2 to 250.2 µM and 99.5 to 192.7 µM, respectively.

## 1. Introduction

The Rutaceae family is a large group of plants with around 150 genera and 1800 species, which are widespread in tropical and temperate regions, especially in Southern Africa and Australia. *Citrus × paradisi* is a *Citrus maxima* backcross or, more precisely, a hybrid of *Citrus maxima* (Burm.) Merr. and *Citrus sinensis* (L.) Osbeck, the latter being a hybrid resulting from a cross between *Citrus maxima* (Burm.) Merr. and *Citrus reticulata* Blanco. It was reported that *Citrus × paradisi* has its origin in the Caribbean, where both parents had been introduced previously [1,2]. From an ethnopharmacological standpoint, the plant is reputed for its use as a local treatment of an array of human diseases. For example, an alcoholic decoction of the seeds is applied against anemia, diabetes mellitus and obesity by some Yoruba herbalists living in Southwest Nigeria [3]. Furthermore, the plant has been used as a folk medicine in many countries as an antibacterial, antifungal, anti-inflammatory, antioxidant, antiviral and preservative. It is also believed to be effective in cancer prevention, cellular regeneration, lowering cholesterol, cleansing, detoxification and heart health maintenance, as well as weight loss, rheumatoid arthritis and inflammation of the kidneys caused by systemic lupus erythematosus. In India, the plant has long been applied for the treatment of anorexia, benign prostatic hypertrophy, prostate-, skin-, colon- and breast cancer, hypercholesterolemia, insomnia and mycosis [4,5].

Previous phytochemical studies carried out on *C. × paradisi* highlighted the presence of volatile constituents in the cold-pressed peel essential oil, including limonene (91.1%), *α*-terpinene (1.3%), α-pinene (0.5%) and, in minor amounts, β-caryophyllene, α-cubebene, (*E*,*E*)-α-farnesene, heptyl acetate, octanal, decanal, citronellal, (*Z*)-carvone, perillene, (*E*)-carveol, perillyl acetate, nootkatone, α- and β-sinensal, methyl-*N*-methylanthranilate and (*Z*,*E*)-farnesol [6,7], as well as some phenolic compounds such as naringin, neohesperidin, hesperidin, neoeriocitrin, nobiletin, gallic acid, chlorogenic acid, caffeic acid and ferulic acid, with naringin being the predominant flavone [8]. Our recent work on the MeOH extract of the stem bark of *C. × paradisi* led to the isolation of a 23*S*-isolimonexic acid derivative, as well as coumarin and acridone alkaloids [9]. Here, as part of our continuous study of the phytochemistry and pharmacology of the genus *Citrus*, we present the results of the chemical and biological investigation of the MeOH extract of the roots of the hybrid species *C. × paradisi*.

## 2. Results and Discussion

The roots of *Citrus × paradisi* Macfad. were extracted with MeOH. The crude extracts were fractionated by liquid/liquid extraction. The fractions were subjected to column chromatography (silica gel) and preparative Thin Layer Chromatography (TLC) to afford two new and ten known compounds. The structures of the known compounds 1,1-dimethylpyrrolidin-1-ium-2-carboxylate (**3**), furan-2,3-diol (**4**), 5-methoxyseselin (**5**), umbelliferone (**6**), scopoletin (**7**), citracridone I (**8**), citracridone II (**9**), citracridone III (**10**), limonin (**11**) and lupeol (**12**) were established using spectroscopic data and previous reports [10,11,12,13] (Figure 1).

Compound **1** was obtained as a yellow powder from hexane-CH_2_Cl_2_ (3/1). The molecular composition was identified by HRESIMS as [(C_10_H_12_NO_2_)_2_Na]^+^ with the [2M+Na]^+^ signal at *m*/*z* 379.16168 (calcd. 379.16173), showing 6 degrees of unsaturation. The presence of hydroxyl and aldehyde functions was indicated by IR bands at λ_max_ 3300 and 1750 cm^−1^, respectively.

The ^1^H-NMR spectrum (Table 1, Appendix A) showed a phenolic signal at *δ*_H_ 11.60 (OH, s), an AB system of two aromatic protons at *δ*_H_ 6.90 (1H, d, *J* = 8.6 Hz) and 7.76 (1H, d, *J* = 8.6 Hz), one aromatic resonance at *δ*_H_ 7.45 (1H, brs), two methylene proton resonances at *δ*_H_ 3.19 (1H, d, *J* = 4.4 Hz) and 3.98 (1H, d, *J* = 4.4 Hz) and an *N*-methyl proton resonance at *δ*_H_ 3.82 (1H, s) (Appendix A). Notably, these signals are characteristic of an *N*-methyl-indoline structure [14]. The *N*-methyl-indoline structure was confirmed by the ^13^C-NMR and DEPT spectra (Table 1, Appendix A), which revealed the presence of 10 carbon atoms, including two methylene carbon resonances at *δ*_C_ 26.2 (C-3) and 57.6 (C-2), an *N*-methyl carbon resonance at *δ*_C_ 50.9 (*N*-CH_3_) and six aromatic carbon resonances at *δ*_C_ 105.7 (C-8), 110.8 (C-7), 125.1 (C-6), 126.8 (C-5), 130.0 (C-9) and 160.7 (C-5). In addition, the ^1^H and ^13^C-NMR spectra presented signals characteristic of the carbonyl of an aldehyde group at *δ*_H_ 9.16 (1H, s) and *δ*_C_ 163.7 (C=O), respectively.

The positions of the hydroxyl and the aldehyde groups were determined by HMBC analysis. The correlation between the aldehyde proton (*δ*_H_* *9.16) and carbons *N*-CH_3_ (*δ*_C_ 50.9), C-2 (*δ*_C_ 57.6) and C-8 (*δ*_C_ 105.7), as well as that between H-2 (*δ*_H_ 3.98) and carbons *N*-CH_3_ (*δ*_C_ 50.9), C-8 (*δ*_C_ 105.7), C-9 (*δ*_C_ 130.0), C=O (*δ*_C_ 163.7) and C-3 (*δ*_C_ 26.2), clearly positioned the aldehyde group at the nitrogen of *N*-methyl-indoline (Figure 2). Finally, the correlation between H-7 (*δ*_H_ 6.90) and carbons C-5 (*δ*_C_ 160.7) and C-9 (*δ_C_* 130.0), as well as that between OH-5 (*δ*_H_ 11.60) and carbons C-6 (*δ*_C_ 125.1) and C-4 (*δ*_C_ 126.8), confirmed the position of the hydroxyl group at C-5. Based on the evidence above, compound **1** was characterized as 1-formyl-5-hydroxy-*N*-methylindolin-1-ium and was given the trivial name paradisinium.

Compound **2** was obtained as a white powder from a fraction eluted with hexane/EtOAc (1/1). The molecular formula was determined by HRESIMS as C_30_H_39_O_9_, giving an [M+H]^+^ ion peak at *m*/*z* 543.25932 (calcd. 543.25937), with 12 degrees of unsaturation. Furthermore, the UV spectrum showed bands at 320 and 260 nm. The presence of hydroxyl, carbonyl and aromatic functions was indicated by IR bands at *υ*_max_ 3450, 1720, 1621 and 1580 cm^−1^. The ^1^H-NMR data (Table 2, Appendix A) indicated the presence of an AB system of two conjugated double bonds at δ_H_ 6.31 (1H, d, *J* = 9.5 Hz, H-3) and δ_H_ 7.67 (1H, d, *J* = 9.5 Hz, H-4) and a singlet at 6.53 (1H, s, H-5). The ^13^C-NMR spectrum (Table 2, Appendix A) showed signals at *δ*_C_ 161.4 (C=O), 146.1 (C-6), 144.4 (C-4), 138.6 (C-9), 137.7 (C-7), 132.1 (C-8), 113.6 (C-3), 111.6 (C-10) and 100.2 (C-5), confirming the presence of a coumarin moiety. Furthermore, the ^1^H-NMR data indicated two symmetric protons at *δ*_H_ 6.64 (2H, s, H-2′/H-6′) and four oxymethine protons at *δ*_H_ 3.66 (1H, dd, *J* = 12.5, 7.0 Hz, H-9a′), 3.76 (1H, dd, *J* = 12.5, 7.0 Hz, H-9b′), 4.10 (1H, m, H-8′) and 4.99 (1H, d, *J* = 13.7 Hz, H-7′), while the ^13^C NMR spectrum showed signals at *δ*_C_ 147.6 (C-3′/C-5′), 135.8 (C-4′), 125.9 (C-1′), 113.6 (C-2′), 104.6 (C-6′), 78.8 (C-8′), 76.8 (C-7′) and 61.2 (C-9′). The data above are consistent with the presence of a phenylpropanoyl moiety, confirming a cleomiscosin skeleton for compound **2** [15,16,17].

In addition, the ^1^H-NMR spectrum showed signals of three methoxy groups at *δ_H_* 3.86 (OCH_3_-6) and 3.84 (OCH_3_-2′/5′) and at *δ_H_* 3.60 (m, H-1″), 2.00 (m, H-2″) 1.21 (brs, H-2″-8″) and 0.83 (t, *J* = 7.5, CH_3_-9″) characteristic of a long chain, which was confirmed by signals at *δ*_C_ 60.7 (C-1″), 31.8 (C-2″), 22.6–29.6 (C-3″-8″) and 14.0 (CH_3_-9″) observed on the ^13^C-NMR spectrum. The long carbon chain was determined by the acid hydrolysis reaction followed by the EI-MS (C_9_H_20_O; [M]^+^; *m*/*z* 144.8) and NMR analysis (500 MHz, CDCl_3_), with signals at δ_H_ 3.67 (CH_2_-OH, t, *J* = 6.73 Hz), 1.58 (HOCH_2_-CH_2_, m), 1.30 ((CH_2_)_n_, brs) and 0.90 (CH_3_, t, *J* = 6.84 Hz). For the linkage between the long chain and the cleomiscosin skeleton, the HMBC spectrum provided the evidence (Figure 3). In this case, the correlation between the methylene proton (*δ_H_ *3.76 and 3.66) and carbons C-1″ (*δ*_C_ 60.7), C-7′ (*δ*_C_ 75.6) and C-8′ (*δ*_C_ 78.8) clearly indicated that this linkage was due to carbon C-9′. The coupling constant between H-7′ and H-8′ was observed to be 13.7 Hz, demonstrating that the two hydrogens were *trans*-oriented [16]. Based on the evidence above, compound **2** was characterized as decyloxycleomiscosin D.

The MeOH extract of the roots and the isolated compounds **1**–**7** were screened against the human adenocarcinoma alveolar basal epithelial cell line A549 and the Caucasian prostate adenocarcinoma cell line PC3 using the MTT assay (Table 3). While the extract showed significant activity, with IC_50_ values of 35.2 ± 2.3 and 38.1 ± 2.5 µg/mL, respectively, compounds **1**–**7** showed weak activity against both cell lines, with IC_50_ values of 99.2 ± 9.6 to 250.2 ±10.2 µM and 99.5 ± 11.5 to 192.7 ± 12.93 µM, respectively (Table 3).

## 3. Materials and Methods

### 3.1. General Experimental Procedures

Ultraviolet spectra were recorded with MeOH on a Hitachi UV 3200 spectrophotometer, (Hitachi, Tokyo, Japan) and infrared spectra were recorded on a JASCO 302-A spectrophotometer (Thermo Scientific, Waltham, MA, USA). ESIMS were measured on an Agilent 6220 TOF LCMS (Santa Clara, CA, USA), and EIMS were measured on a Finnigan MAT 95 spectrometer (70 eV) (Thermo Fisher Scientific, Darmstadt, Germany), with perfluorokerosene as the reference substance for HRESIMS. The ^1^H-NMR and ^13^C-NMR spectra were recorded at 500 MHz and 125 MHz, respectively, on a Bruker DRX 500 NMR spectrometer (Bruker, Rheinstetten, Germany). The chemical shifts are reported in *δ* (ppm) using tetramethylsilane (TMS) (Sigma-Aldrich, Munich, Germany) as the internal standard, while the coupling constants (*J*) were measured in Hz. Column chromatography was carried out on silica gel 230–400 and silica gel 70–230 mesh (Merck, Darmstadt, Germany). Thin-layer chromatography (TLC) was performed on precoated silica gel 60 F_254_ aluminum foil (Merck, Darmstadt, Germany), and spots were detected using diluted sulfuric acid spray reagent after heating the chromatogram. The degree of purity of the positive control compounds was ≥ 98%, while that of the isolated compounds was > 95%. Doxorubicin was purchased from Sigma-Aldrich (Germany). The Caucasian prostate adenocarcinoma cell line PC-3 (CRL-1435) and the human adenocarcinoma alveolar basal epithelial A-549 (CCL-185) were purchased from the American Type Culture Collection (ATCC), 10,801 University Blvd., Manassas, USA. All reagents used were of analytical grade.

### 3.2. Plant Material

In December 2018, the roots of *Citrus × paradisi* were collected at Penja, a locality in the Littoral region of Cameroon. At the National Herbarium, the botanist Ngansop identified the species and arranged a deposit under the voucher number HNC 67471/67471.

### 3.3. Extraction and Isolation

A total of 3.2 kg of air-dried powdered roots of *Citrus × paradisi* were extracted with MeOH (5.0 L) at room temperature over 3 days. The brown extract was filtered and concentrated under reduced pressure to yield 35.5 g. A total of 33.0 g of the extract was subjected to flash column chromatography on silica gel (70–230 mesh) and eluted with *n*-hexane, using mixtures of *n*-hexane/EtOAc and EtOAc/MeOH of increasing polarities. A total of 85 fractions of 200 mL each were collected and combined on the basis of their similar TLC profiles to yield 4 main fractions: F1 (5.2 g), F2 (7.5 g), F3 (8.4 g) and F4 (10.1 g).

Fraction F1 (5.2 g) was further chromatographed using a silica gel column with an *n*-hexane-EtOAc gradient. A total of 25 fractions of approximately 100 mL each were collected and combined on the basis of TLC. Subfractions 1–15 were eluted with a mixture of *n*-hexane/EtOAc (8.5:1.5) to yield lupeol (**12**) (15.1 mg), umbelliferone (**6**) (13.2 mg) and 5-methoxyseselin (**5**) (7.5 mg). F2 (7.5 g) was also subjected to column chromatography on silica gel (70–230 mesh; Merck) and eluted with *n*-hexane/EtOAc (3:1–1:3). Subfractions 26–35 yielded scopoletin (**7**) (9.8 mg), while subfractions 39–60 afforded furan-2,3-diol (**4**) (14.5 mg) and decyloxycleomiscosin D (**2**) (11.0 mg). F3 (8.4 g) was subjected to column chromatography on silica gel (70–230 mesh; Merck) and eluted with *n*-hexane/EtOAc mixtures (1:1–1:4). Subfractions 61–75 yielded citracridone I (**8**) (21.5 mg), citracridone II (**9**) (5.2 mg) and citracridone III (**10**) (11.5 mg). Finally, fraction F4 (10.1 g) was chromatographed using a silica gel (70–230 mesh; Merck) column with an *n*-hexane/EtOAc gradient. A total of 55 fractions of approximately 100 mL each were collected and combined on the basis of TLC. Subfractions 10–55 were eluted with EtOAc and a mixture of EtOAc/MeOH (19.5/0.5), delivering 1-formyl-5-hydroxy-*N*-methylindolin-1-ium (**1**) (10.3 mg), dimethylpyrrolidin-1-ium-2-carboxylate (**3**) (10.5 mg) and limonin (**11**) (21.3 mg).

#### 3.3.1. 1-Formyl-5-hydroxy-*N*-methylindolin-1-ium (**1**)

Yellow powder (MeOH); R_f_ = 0.30 (CH_2_Cl_2_/MeOH; 5/1); IR (KBr); λ_max_ 3300, 2925, 2700, 1750, 1620, 1000, 750 cm^−1^; ^1^H-NMR (500 MHz, DMSO-d_6_) and ^13^C-NMR (125 MHz, DMSO-*d*_6_) data, see Table 1; HR-ESIMS [2M+Na]^+^ at *m*/*z* = 379.16168, (calcd. 379.16173) for [(C_10_H_12_NO_2_)_2_Na]^+^.

#### 3.3.2. Decyloxycleomiscosin D (**2**)

White powder (MeOH); R_f_ = 0.55 (CH_2_Cl_2_/MeOH; 5/1); IR (KBr); λ_max_ 3450, 2900, 2750, 1720, 1621, 1580, 750 cm^−1^; UV (CH_3_OH): *ʋ*_max_ 320, 260 nm; ^1^H-NMR (500 MHz, CHCl_3_/CD_3_OD; 10/1) and ^13^C-NMR (125 MHz, CHCl_3_/CD_3_OD; 10/1) data, see Table 2; HRESIMS [M+H]^+^ at *m*/*z* 543.25932 (calcd. 543.25941) for C_30_H_39_O_9_. The long carbon chain was determined by the acid hydrolysis reaction of ethers. A total of 2 mg of compound **2** was dissolved in an HCl (35% *v*/*v*)/H_2_O mixture at 60 °C for 20 min and then neutralized with diluted KOH. A total of 0.25 mg of nonanol was obtained, with a yield of 47.2%.

### 3.4. Cytotoxicity Assay

The cytotoxic activity of the MeOH root extract of *C. × paradisi* and compounds **1–7** was assayed against the human adenocarcinoma alveolar basal epithelial A549 and Caucasian prostate adenocarcinoma PC3 cell lines, applying the 3-(4,5-dimethylthiazol-2-yl)-2,5-diphenyltetrazolium bromide (MTT) assay following a protocol which was described previously [18]. In short, the cell suspensions were freshly trypsinized and introduced into 96-well microtiter plates at a density of 1 × 104 cells per well. From stocks diluted in dimethylsulfoxid, the test compounds were added to the wells. After two days of incubation, MTT was added, and the attached cells were solubilized in dichloromethane. When measuring the absorbance at 550 nm with a plate reader, the 50% inhibition concentration (IC50) of cell growth was recorded and compared with the positive control, doxorubicin.

## 4. Conclusions

The chemical investigation of *C. × paradisi* led to the isolation of two new compounds, 1-formyl-5-hydroxy-*N*-methylindolin-1-ium (**1**) and decyloxycleomiscosin D (**2**), along with ten known compounds, namely 1,1-dimethylpyrrolidin-1-ium-2-carboxylate (**3**), furan-2,3-diol (**4**), 5-methoxyseselin (**5**), umbelliferone (**6**), scopoletin (**7**), citracridone I (**8**), citracridone II (**9**), citracridone III (**10**), limonin (**11**) and lupeol (**12**). In addition to the coumarins, acridone alkaloids, limonoids and triterpenes generally encountered in plants of the *Citrus* genus, the small molecules 1,1-dimethylpyrrolidin-1-ium-2-carboxylate (**3**) and furan-2,3-diol (**4**) were isolated here for the first time. The presence of such compounds in this plant may be due to genetic variation as a consequences of plant hybridization. While the methanolic root extract displayed significant activity in the anticancer assays, compounds **1**–**7** showed only weak activity against the human adenocarcinoma alveolar basal epithelial cell line A549 and the Caucasian prostate adenocarcinoma cell line PC3 using the MTT assay. Further research is required to determine whether the activity of the crude extract is reflected by other secondary metabolites or synergistic in nature.

## Figures and Tables

**Figure 1 molecules-28-01078-f001:**
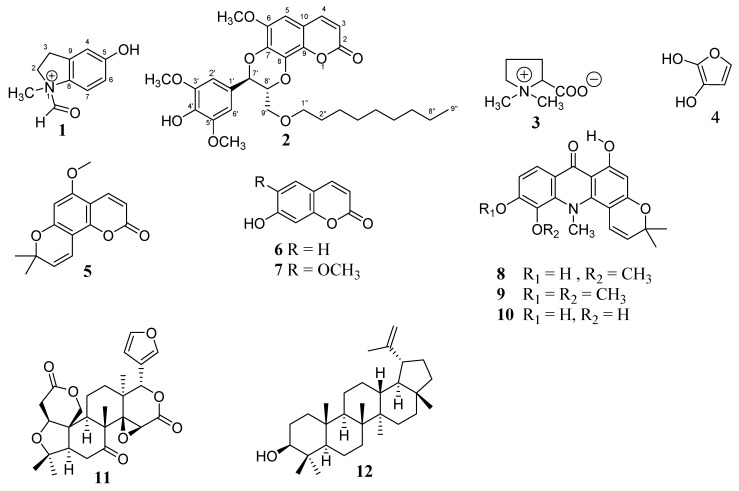
Structures of compounds **1**–**12**.

**Figure 2 molecules-28-01078-f002:**
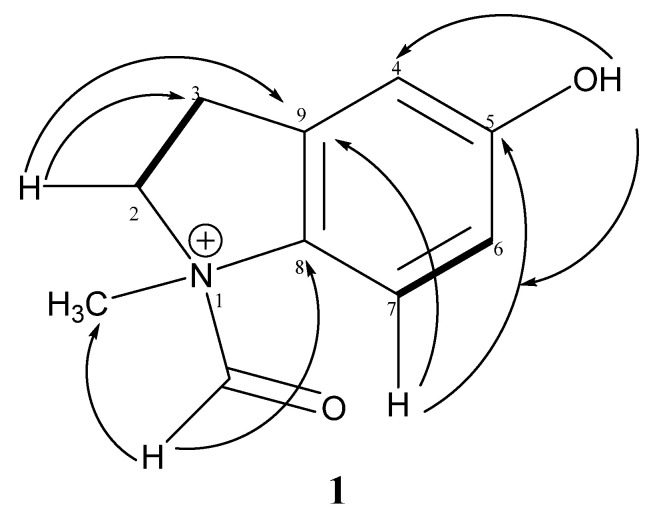
Some correlations of compound **1** HMBC 
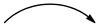
 COSY 

.

**Figure 3 molecules-28-01078-f003:**
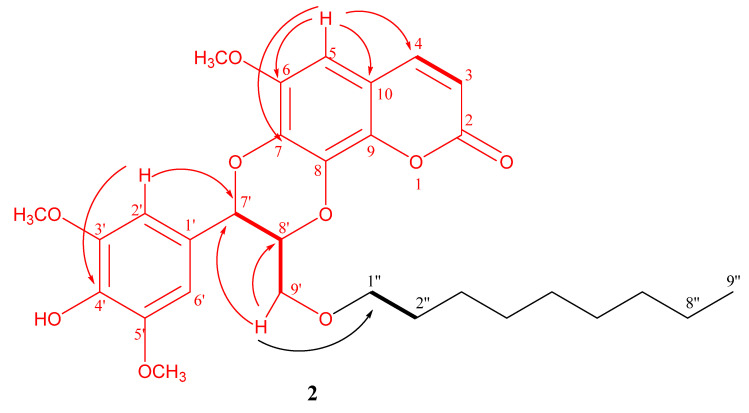
Some correlations of compound **2** HMBC 
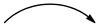
 COSY 

.

**Table 1 molecules-28-01078-t001:** ^1^H (500 MHz) and ^13^C (125 MHz) data of 1-formyl-5-hydroxy-*N*-methylindolin-1-ium (**1**) in DMSO-d_6_.

Attribution	^1^H [m, *J* (Hz)]	^13^C
**1**	-	-
**2**	3.98 (d, 4.4)	57.6
**3**	3.19 (d, 4.4)	26.2
**4**	7.45 (brs)	126.8
**5**	-	160.7
**6**	7.76 (d, 8.6)	125.1
**7**	6.90 (d, 8.6)	110.8
**8**	-	105.7
**9**	-	130.0
**N-CH_3_**	3.82 (s)	50.9
**N-CHO**	9.16 (s)	163.7
**OH**	11.60 (s)	-

Assignments were based on COSY, HSQC, HMBC, and NOESY (Appendix A) experiments.

**Table 2 molecules-28-01078-t002:** ^1^H (500 MHz) and ^13^C (125 MHz) data of decyloxycleomiscosin D (**2**) in CDCl_3_:CD_3_OD (10:1).

Attribution	^1^H [m, *J* (Hz)]	^13^C
**1**	-	-
**2**	-	161.4
**3**	6.31 (d, 9.5)	113.6
**4**	7.67 (d, 9.5)	144.4
**5**	6.53 (s)	100.2
**6**		146.1
**7**		137.7
**8**		132.1
**9**		138.6
**10**		111.6
**1′**		125.9
**2′**	6.64 (s)	113.6
**3′**		147.6
**4′**		135.8
**5′**		147.6
**6′**	6.64 (s)	104.6
**7′**	4.99 (d,13.7)	75.6
**8′**	4.10 (m)	78.8
**9′**	3.66 (dd, 12.5; 6.9) 3.76 (dd, 12.5; 7.0)	61.2
**1″**	3.60 (m)	60.7
**2″**	2.00 (m)	31.8
**3″–8″**	1.21 (brs)	22.6–29.6
**9″**	0.83 (t; 7.5)	14.0
**MeO-6**	3.86 (s)	56.4
**MeO-3′**	3.84 (s)	56.3
**MeO-5′**	3.84 (s)	56.3

Assignments were based on COSY, HSQC, HMBC and NOESY experiments Appendix A).

**Table 3 molecules-28-01078-t003:** Cell growth inhibitory activities.

Sample	A549	PC-3
**1**	112.5 ± 11.5 µM	99.5 ± 11.5 µM
**2**	250.2 ± 10.2 µM	192.7 ± 12.3 µM
**3**	99.2 ± 9.5 µM	100.2 ± 12.7 µM
**4**	152.5 ± 11.3 µM	147.3 ± 21.5 µM
**5**	201.3 ± 13.5 µM	180.2 ± 14.3 µM
**6**	156.2 ± 18.2 µM	150.2 ± 21.3 µM
**7**	135.6 ± 21.8 µM	1335 ± 22.6 µM
root extract	35.2 ± 2.3 µg/mL	38.1 ± 2.5 µg/mL
doxorubicin	0.9 ± 0.1 µM	1.6 ± 0.2 µM

Data are represented as mean ± SEM (Standard Error of the Mean) (*n* = 3); IC_50_ = sample concentration that caused 50% cell growth inhibition.

## Data Availability

Not applicable.

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
