# Peer review of "Coumarinolignoid and Indole Alkaloids from the Roots of the Hybrid Plant Citrus × paradisi Macfad (Rutaceae)"

_molecules, 2023, doi:10.3390/molecules28031078_

Round 1
Reviewer 1 Report
Summary
In this manuscript, the authors perform the isolation of the natural products from the roots of Citrus x paradisi Macfad. (Rutaceae). A total of 12 compounds where identified, for which two are reported for the first time. I think that this article would be of interest to the community. However, I would suggest that it can be accepted for publication after some modifications.
Major issues
Line 41 - "From an ethnopharmacological standpoint, the plant is reputed for the local management of an array of human diseases." I think that the use of "local management" might be misleading. Local can be confused with topical route of administration. Management is a strong term that brings the idea that several diseases can be treated with this plant. The plant has gained a reputation for alleviating some symptoms or to aid on the prevention of diseases.
Line 50 - "Locally, the plant has long been applied for the treatment of anorexia, benign prostatic 50 hypertrophy, prostate-, skin-, colon- and breast cancer, hypercholesterolemia, insomnia 51 and mycosis [3,4]." In accordance with the sentence on line 44, the authors should specify where the plant has been applied (country or region).
line 60 - "Our recent work on the MeOH extract of the stem bark of C. x paradisi led to the isolation of a 23S-isolimonexic acid derivative, as well as coumarine and acridone alkaloids [8]". What are the differences and similarities between the two extracts? The authors should describe the comparison between this work and their previous work in the conclusions.
line 96 and 195: The authors mention that the spectrum was recorded with CD3OD, but the NMR spectra displayed in the Supplementary Materials shows the solvent residual peak of DMSO-d6.
line 114 - "The 1H NMR data indicated the presence of a coumarin moiety by resonances at δH 6.31 and 7.67 (lH, d each, J = 9.5 Hz) and 6.53 (1H, s, H-5), a phenylpropanoyl moiety at δH 6.64 (2H, s, H-2′/H-6′)." I understand what the authors are trying to say, but the presence of a signal at δH 6.64 (2H) per si does not indicate the presence of a coumarin or a phenylpropanoyl moiety. The authors should mention in the same sentence the 1H and 13C NMR data that indicates the presence of a coumarin moiety and the phenylpropanoyl moiety.
Suggestions
To confirm the NMR data described in tables 1 and 2, the reader must cross it with figure 1. The figure and tables are on different pages. I would suggest dividing figure 2 into two figures to place them (with attribution number on the structures) nearby Table 1 and Table 2.
line 122: The information above confirmed a cleomiscosin skeleton for compound 2 [14–16]. I would suggest highlighting with a different color the cleomiscosin skeleton on the decyloxycleomiscosin D structure presented in figure 2.
Minor issues
line 42: Missing reference in this sentence: "For example, an alcoholic decoction of the seeds is applied against anemia, diabetes mellitus and obesity by some Yoruba herbalists living in Southwest Nigeria."
line 114: 1 instead of l: "(lH, d each, J = 9.5 Hz)"
line 136: "In the following" is not needed in this sentence: "In the following, the MeOH extract of the roots and the isolated compounds 1–7 were screened against the human adenocarcinoma alveolar basal epithelial cell line A549 and the Caucasian prostate adenocarcinoma cell line PC3 using the MTT assay (Table 3)."
line 242: Supplementary Materials description is incomplete.
Supplementary Materials: Revise captions. Figure S1 does not have a caption. Figure S2 and Figure S11 captions are on the following page.
Author Response
The authors thank the reviewers for the reading, all thier comments and remarks to improve the quality of the manuscript before acceptance and publication. We present here below our answer (in red) to the questions and remarks of the reviewer. The authors remain with great hope that their answers will meet the requirements of the reviewer and the editor too.
Summary
In this manuscript, the authors perform the isolation of the natural products from the roots of Citrus x paradisi Macfad. (Rutaceae). A total of 12 compounds where identified, for which two are reported for the first time. I think that this article would be of interest to the community. However, I would suggest that it can be accepted for publication after some modifications.
Major issues
Line 41 - "From an ethnopharmacological standpoint, the plant is reputed for the local management of an array of human diseases." I think that the use of "local management" might be misleading. Local can be confused with topical route of administration. Management is a strong term that brings the idea that several diseases can be treated with this plant. The plant has gained a reputation for alleviating some symptoms or to aid on the prevention of diseases.
We have corrected the word management by treatment.
Line 50 - "Locally, the plant has long been applied for the treatment of anorexia, benign prostatic 50 hypertrophy, prostate-, skin-, colon- and breast cancer, hypercholesterolemia, insomnia 51 and mycosis [3,4]." In accordance with the sentence on line 44, the authors should specify where the plant has been applied (country or region).
Corrected
line 60 - "Our recent work on the MeOH extract of the stem bark of C. x paradisi led to the isolation of a 23S-isolimonexic acid derivative, as well as coumarine and acridone alkaloids [8]". What are the differences and similarities between the two extracts? The authors should describe the comparison between this work and their previous work in the conclusions.
The major difference between the two extracts lies in the part of the harvest (the bark of the trunk for one and the roots for the second) in addition the extraction technique applied to the second extract made it possible to isolate more acridones and flavonoids, small molecules.
line 96 and 195: The authors mention that the spectrum was recorded with CD3OD, but the NMR spectra displayed in the Supplementary Materials shows the solvent residual peak of DMSO-d6.
Corrected (DMSO-d6)
line 114 - "The 1H NMR data indicated the presence of a coumarin moiety by resonances at δH 6.31 and 7.67 (lH, d each, J = 9.5 Hz) and 6.53 (1H, s, H-5), a phenylpropanoyl moiety at δH 6.64 (2H, s, H-2′/H-6′)." I understand what the authors are trying to say, but the presence of a signal at δH 6.64 (2H) per si does not indicate the presence of a coumarin or a phenylpropanoyl moiety. The authors should mention in the same sentence the 1H and 13C NMR data that indicates the presence of a coumarin moiety and the phenylpropanoyl moiety.
We have reformulated to make the text more understandable
Suggestions
To confirm the NMR data described in tables 1 and 2, the reader must cross it with figure 1. The figure and tables are on different pages. I would suggest dividing figure 2 into two figures to place them (with attribution number on the structures) nearby Table 1 and Table 2.
Corrected in the text
line 122: The information above confirmed a cleomiscosin skeleton for compound 2 [14–16]. I would suggest highlighting with a different color the cleomiscosin skeleton on the decyloxycleomiscosin D structure presented in figure 2.
Corrected in the text
Minor issues
line 42: Missing reference in this sentence: "For example, an alcoholic decoction of the seeds is applied against anemia, diabetes mellitus and obesity by some Yoruba herbalists living in Southwest Nigeria."
Corrected
line 114: 1 instead of l: "(lH, d each, J = 9.5 Hz)" Corrected
line 136: "In the following" is not needed in this sentence: "In the following, the MeOH extract of the roots and the isolated compounds 1–7 were screened against the human adenocarcinoma alveolar basal epithelial cell line A549 and the Caucasian prostate adenocarcinoma cell line PC3 using the MTT assay (Table 3)." Corrected
line 242: Supplementary Materials description is incomplete. Corrected
Supplementary Materials: Revise captions. Figure S1 does not have a caption. Figure S2 and Figure S11 captions are on the following page. Corrected
Reviewer 2 Report
The submitted manuscript ‘Coumarinolignoid and indole alkaloids from the roots of the hybrid plant Citrus x paradisi Macfad. (Rutaceae)’ by Sewald and Wansi together with their coworkers reported a detailed chemical investigation of the root extract of Citrus x paradisi. As a result, twelve structurally diverse compounds including two new ones were discovered. Moreover, the extract and compounds 8–12 exhibited different levels of cytotoxicity against cell lines cell line A549 and PC3. This work is important, since the findings give an insight of the chemical constituents of C. x paradisi for its cancer prevention efficacy.
However, I will recommend this manuscript for acceptance after the following concerns are revised.
1. According to Table 3, the IC50 values of extract were 35.2 and 38.1 μg/mL not 35.2 and 38.1 μM. Please revise them in the Abstract and the main text.
2. As the new compound 1 must be isolated for the first time in this study, please delete this compound in the corresponding sentence in the Conclusions part.
3. The mass peaks at ESI spectra (Figures S1 and S9) were not consistent with those mentioned in the manuscript on P2 and P4. Please provide the HRESIMS spectra of the two compounds 1 and 2.
4. What MR solvent was used for compound 2? According to the 1H and 13C NMR spectra (Figures S10 and S12), the NMR solvent was a mixture of CDCl3 and CD3OD. If so, please give the ratio of CDCl3:CD3OD in the Table 2 caption and revised the NMR solvent recorded in the Materials and Methods part.
5. The term ‘a phenylpropanoyl moiety’ was not proper for the left benzene ring of the structure of compound 2.
6. Please give a description to determine the configuration of C-7ꞌ and C-8ꞌ of compound 2.
7. Why were compounds 8–12 not evaluated in the cytotoxicity assay? The significant cytotoxicity of the extract might be attributed to these compounds.
Others:
1. P2L73: Incoherent serial numbers of references ‘[9,12]’. Perhaps it was ‘[9–12]’.
2. P4L114: ‘δH 6.31 and 7.67 (lH, d each, J = 9.5 Hz)’ → ‘δH 6.31 and 7.67 (each 1H, d, J = 9.5 Hz)’
3. P4L129: Please revise the sentence ‘To link the long chain to the cleomiscosin skeleton, the HMBC spectrum was used.’ as ‘For the linkage between the long chain and the cleomiscosin skeleton, the HMBC spectrum gave the evidence.’
Author Response
The authors thank the reviewers for the reading, all thier comments and remarks to improve the quality of the manuscript before acceptance and publication. We present here below our answer (in red) to the questions and remarks of the reviewer. The authors remain with great hope that their answers will meet the requirements of the reviewer and the editor too.
The submitted manuscript ‘Coumarinolignoid and indole alkaloids from the roots of the hybrid plant Citrus x paradisi Macfad. (Rutaceae)’ by Sewald and Wansi together with their coworkers reported a detailed chemical investigation of the root extract of Citrus x paradisi. As a result, twelve structurally diverse compounds including two new ones were discovered. Moreover, the extract and compounds 8–12 exhibited different levels of cytotoxicity against cell lines cell line A549 and PC3. This work is important, since the findings give an insight of the chemical constituents of C. x paradisi for its cancer prevention efficacy.
However, I will recommend this manuscript for acceptance after the following concerns are revised.
- According to Table 3, the IC50 values of extract were 35.2 and 38.1 μg/mL not 35.2 and 38.1 μM. Please revise them in the Abstract and the main text. Corrected
- As the new compound 1 must be isolated for the first time in this study, please delete this compound in the corresponding sentence in the Conclusions part. Corrected
- The mass peaks at ESI spectra (Figures S1 and S9) were not consistent with those mentioned in the manuscript on P2 and P4. Please provide the HRESIMS spectra of the two compounds 1 and 2.
We did not receive the HRESIMS of the two compounds, but only the values from the HRESIMS and the corresponding molecular formula from the operator. Normally the values of exact mass giving by the HR is always different from the ESI simple mass.
- What MR solvent was used for compound 2? According to the 1H and 13C NMR spectra (Figures S10 and S12), the NMR solvent was a mixture of CDCl3 and CD3OD. If so, please give the ratio of CDCl3:CD3OD in the Table 2 caption and revised the NMR solvent recorded in the Materials and Methods part. Corrected
- The term ‘a phenylpropanoyl moiety’ was not proper for the left benzene ring of the structure of compound 2.
We used the name phenylpropanoyl to facilitate the description of compound 2, because we decided to divide the molecule in two parts, Coumarin and lignane (phenylpropanoyl).
- Please give a description to determine the configuration of C-7ꞌ and C-8ꞌ of compound 2. Added and Corrected
7. Why were compounds 8–12 not evaluated in the cytotoxicity assay? The significant cytotoxicity of the extract might be attributed to these compounds. We carried out the tests of the compounds which had not been tested before
Others:
- P2L73: Incoherent serial numbers of references ‘[9,12]’. Perhaps it was ‘[9–12]’. Corrected
- P4L114: ‘δH 6.31 and 7.67 (lH, d each, J = 9.5 Hz)’ → ‘δH 6.31 and 7.67 (each 1H, d, J = 9.5 Hz)’ Corrected
- P4L129: Please revise the sentence ‘To link the long chain to the cleomiscosin skeleton, the HMBC spectrum was used.’ as ‘For the linkage between the long chain and the cleomiscosin skeleton, the HMBC spectrum gave the evidence.’ Corrected
Round 2
Reviewer 2 Report
Authors forgot to add the ratio 10:1 of CDCl3:CD3OD in the Table 2 caption. However, the resubmitted manuscript has been improved and it is recommended to accept it